# Efficacy and Safety of Plasma Rich in Growth Factor in Patients with Congenital Aniridia and Dry Eye Disease

**DOI:** 10.3390/diseases12040076

**Published:** 2024-04-11

**Authors:** Javier Lozano-Sanroma, Alberto Barros, Ignacio Alcalde, Rosa Alvarado-Villacorta, Ronald M. Sánchez-Ávila, Juan Queiruga-Piñeiro, Luis Fernández-Vega Cueto, Eduardo Anitua, Jesús Merayo-Lloves

**Affiliations:** 1Instituto Universitario Fernández-Vega, Fundación de Investigación Oftalmológica, 33012 Oviedo, Spain; alberto.barros@fernandez-vega.com (A.B.); nacho.alcalde@fio.as (I.A.); juan.queiruga@fernandez-vega.com (J.Q.-P.); merayo@fio.as (J.M.-L.); 2Instituto de Investigación Sanitaria del Principado de Asturias (ISPA), Universidad de Oviedo, 33011 Oviedo, Spain; 3Regenerative Medicine Laboratory, Biotechnology Institute (BTI), 01007 Vitoria, Spain; rsanchez@bti-health.com (R.M.S.-Á.); eduardo@fundacioneduardoanitua.org (E.A.)

**Keywords:** congenital aniridia, plasma rich in growth factors, PRGF, ocular redness, DED

## Abstract

Congenital aniridia is a rare bilateral ocular malformation characterized by the partial or complete absence of the iris and is frequently associated with various anomalies, including keratopathy, cataract, glaucoma, and foveal and optic nerve hypoplasia. Additionally, nearly 50% of individuals with congenital aniridia experience symptoms of ocular dryness. Traditional treatment encompasses artificial tears and autologous serum. This study aimed to assess the effectiveness and safety of using platelet rich in growth factors (PRGF) plasma in patients with congenital aniridia and ocular dryness symptoms. Methods: The included patients underwent two cycles of a 3-month PRGF treatment. At 6 months, symptomatology was evaluated using the OSDI and SANDE questionnaires, and ocular surface parameters were analyzed. Results: The OSDI and SANDE values for frequency and severity demonstrated statistically significant improvements (*p* < 0.05). Ocular redness, corneal damage (corneal staining), and tear volume (Schirmer test) also exhibited statistically significant improvements (*p* < 0.05). No significant changes were observed in visual acuity or in the grade of meibomian gland loss. Conclusion: The use of PRGF in patients with congenital aniridia and ocular dryness symptoms led to significant improvements in symptomatology, ocular redness, and ocular damage. No adverse effects were observed during the use of PRGF.

## 1. Introduction

Isolated aniridia (i.e., without systemic involvement) is a congenital bilateral ocular malformation characterized by the partial or complete absence of the iris. It is classified by the European Consortium for Rare Diseases under the code ORPHA: 250923. Its estimated prevalence varies from 1 to 9 per 100,000 people depending on the population and country studied [1,2,3]. 

Aniridia primarily occurs due to mutations in the *PAX6* gene (11p13), which encodes a transcriptional regulator involved in ocular development. Mutations in *PAX6* lead to alterations in corneal cytokeratin expression, cell adhesion, and glycoconjugate expression [4]. Approximately 60% of cases have a familial inheritance pattern, while one-third of cases appear sporadically [5]. Diagnosis is based on ophthalmological examination and confirmed by detecting mutations in the *PAX6* gene. It is also important to rule out variants, leading to the Wilms tumor gene (*WT1*) or WAGR syndrome, which includes the Wilms tumor, aniridia, genitourinary anomalies, and mental retardation. Hereditary aniridia is transmitted in an autosomal dominant manner with high penetrance and variable expression [4].

In addition to iris hypoplasia, a variety of other ocular anomalies can occur, including cataracts, glaucoma, aniridia-associated keratopathy (AAK), optic nerve or foveal hypoplasia, nystagmus, photophobia, and dry eye disease (DED), all of which result in varying degrees of vision impairment. The average visual acuity is around 0.19 (decimal scale) in young adults according to ORPHA. These data align with a study of patients in Sweden and Norway [6].

Of the aforementioned complications, AAK, photophobia, and dry eye, in addition to visual impairment, cause discomfort to patients. For photophobia, the use of glasses with a protective filter, which reduces visible light, is recommended. Surgical techniques, such as limbal stem cell transplantation through a keratolimbal allograft, have been evaluated as treatments for AAK, which is one of the leading causes of vision loss in patients with aniridia. Other surgical procedures tend to fail unless there is an approach involving limbal stem cells [1,7,8,9,10,11].

In less severe cases of AAK, topical treatment is recommended, such as lubricants in the form of artificial tears or autologous serum (AS) [3]. AS is a biological substitute, and its use for DED dates back to the 1970s [12]. Patients who used AS have reported reduced symptoms, including less tearing, photophobia, and a decreased sensation of a foreign body. The stability of the tear film improved, as did corneal epithelialization. Additionally, a decrease in corneal epithelial squamous metaplasia was observed [13].

Another recently employed blood derivative is plasma rich in growth factors (PRGF). There are numerous preparation methods, specifically up to 30, according to the existing literature [14]. One of these methods is PRGF Endoret^®^, which is a methodologically standardized preparation method [15,16]. PRGF provides a benefit over AS by removing inflammatory factors released by leukocytes, such as IL-6, IL-1B, and TNF-α, which are linked to inflammatory processes in specific ocular conditions [17,18,19]. Consequently, PRGF is a serum enriched with growth factors from platelets, with a concentration that can be up to twice that of AS. Upon activation, it is capable of secreting PDGF, TGF-β, VEGF, and IGF-I. As it originates from the patient’s own body, the potential for immunological reactions is minimized [20].

The utilization of PRGF in the treatment of DED has been previously investigated with positive outcomes observed [21,22,23,24,25,26].

As observed in the literature, at least half of the patients with aniridia experience dry eye symptoms [27], which might be attributed to poor tear quality, indicating a predominantly evaporative DED [2]. This results in low fluorescein tear break-up time (FBUT) values and could be justified by the high prevalence of Meibomian Gland Dysfunction (MGD). This DED could also contribute to the exacerbation of AAK [9].

To date, there have been no published studies on the use of PRGF in patients with aniridia and DED. The purpose of this study was to investigate the effects of employing topical PRGF on the ocular surface of patients with aniridia while assessing their subjective responses.

## 2. Material and Methods

This observational, prospective study was conducted at the Ocular Surface Unit of the Instituto Oftalmológico Fernández-Vega (IOFV) in Oviedo, Principality of Asturias, Spain, spanning from May 2022 to February 2023. Informed consent was obtained from all participating patients, and the study received approval from the Research Ethics Committee of the Principality of Asturias under the reference number CEImPA 2022.113 on 3 May 2022. This study adhered to the principles outlined in the Declaration of Helsinki.

### 2.1. Patients

Participants in the aniridia group were recruited through the patient organization Asociación Española de Aniridia. All individuals included in this study were diagnosed with aniridia and reported experiencing ocular dryness. All participants were above the age of 18 and willingly agreed to take part in this study. They underwent a comprehensive ophthalmological and optometric examination. As exclusion criteria, active infectious processes, platelet disorders, or coagulation abnormalities were considered. All patients were free to leave the study at any time without affecting the therapeutic indication.

### 2.2. Treatment

All patients were administered PRGF four times daily for a duration of six months. No modifications were made to any prior treatments.

#### PRGF Preparation

After obtaining signed informed consent forms, blood samples were collected from the patients and placed into 9 mL tubes. The blood samples for PRGF were then centrifuged at 580× *g* for 8 min at room temperature using an Endoret System centrifuge (BTI Biotechnology Institute, S.L., Miñano, Álava, Spain). The entirety of the plasma column located above the buffy coat was collected using an Endoret ophthalmology kit (BTI Biotechnology Institute, S.L., Miñano, Álava, Spain), with careful avoidance of the layer containing leukocytes. Subsequently, the plasma preparations were subjected to incubation with an Endoret activator (BTI Biotechnology Institute, S.L., Miñano, Alava, Spain) at 37 °C for a minimum of 40 min, followed by further incubation at 56 °C for 30 to 60 min. The resulting PRGF supernatants were filtered, divided into aliquots, and stored at a temperature of −80 °C until ready for use. All of these procedures were conducted under strictly sterile conditions within a laminar flow hood.

Prior to starting treatment, patients were instructed to store the PRGF eye drop dispensers at −20 °C for a maximum of three months. Each dispenser was intended for use over a span of three consecutive days [15,26,28].

### 2.3. Clinical Tests

#### 2.3.1. Ocular Surface Symptom Assessment

To evaluate the symptoms reported by the patients, the Ocular Surface Disease Index (OSDI) was utilized. The OSDI is a 12-item questionnaire [29] that gauges discomfort, visual impairment, and triggers related to the environment. For analysis, the OSDI formula was applied, which is calculated as the sum of scores multiplied by 25, divided by the number of responses.

Additionally, the SANDE (Symptoms Analysis in Dry Eye) scale was used to measure the severity and frequency of symptoms [30]. Patients were required to note their symptoms on a scale ranging from 0 to 100 mm.

#### 2.3.2. Visual Acuity

The Best Corrected Visual Acuity (BCVA) was measured on a decimal scale following retinoscopy. In cases where subjects were unable to discern letters on the optotypes, visual acuity was assessed using a finger-counting method and then converted to a decimal scale.

#### 2.3.3. Ocular Redness

To evaluate inflammation on the ocular surface, the “Oculus Index” integrated into the Keratograph 5M device (Oculus^®^, Wetzlar, Germany) was employed. The system automatically generated a bulbar redness score ranging from 0.0 to 4.0. This score was determined based on the percentage ratio between the vessels and the rest of the analyzed area [31,32,33,34].

#### 2.3.4. Fluorescein Staining

To assess ocular surface damage, a drop of fluorescein (minims fluorescein sodium 20 mg/mL eye drop solution, Laboratoire Chauvin Z.I. Ripotier 07200 Aubenas, France) was administered and applied to the far temporal region of the eye while looking upward to prevent harm to the conjunctival or corneal tissues. The Oxford grading scheme was employed.

#### 2.3.5. Presence and Level of AAK

The presence and severity of aniridia-associated keratopathy were evaluated on a scale of 0 to 4 as per the classification by Lagali et al. [35].

#### 2.3.6. Stability of the Tear Film

The stability of the tear film was assessed using the fluorescein break-up time (FBUT), which denotes the duration, in seconds, before the break-up of the fluorescein-stained tear pattern is observed [36].

#### 2.3.7. Tear Volume

Tear film volume was assessed via the Schirmer test under topical anesthesia. The tip of the strip was gently inserted into the lower temporal canthus, with the lower eyelid slightly retracted toward the temporal side and the gaze directed upwards. This precautionary measure was taken to avoid any potential harm to the conjunctiva, cornea, or eyelid margin. The strip was then left in position for 5 min, and the length of wetting, in millimeters, was recorded [37,38].

#### 2.3.8. Meibography

To study the posterior palpebral aspect, meibography of the upper and lower eyelids was conducted using infrared image capture and digital image enhancement. The Heiko Pult scale was used to evaluate the area of loss, which ranges from 0 (no loss) to 4 (greater than 75% area of loss) [32].

#### 2.3.9. Impression Cytology

Strips of cellulose acetate filter membrane (Sartorius Stedim Biotech GmbH, Göttingen, Germany) were applied to the inferior temporal conjunctiva of patients, inducing a gentle pressure for 10 s. The membranes were then collected and transferred to 2 mL tubes containing 70% ethanol for 10 min for fixation. PAS (Periodic acid of Schiff) staining was performed to evaluate the pathological state of the conjunctiva epithelium. The samples were kept in labeled histological cassettes and immersed for 10 min in 1% periodic acid followed by an additional 10 min in Schiff reagent (both from Merck KGaA, Darmstadt, Germany). Nuclei were counterstained with Harris Hematoxylin for 10 min (Merck). Finally, membranes were mounted on microscope slides and examined under a Leica DM6000 microscope (Leica Microsystems GmbH, Wetzlar, Germany). Goblet cells and epithelial cells were evaluated compared to 3 control preparations from young healthy volunteers.

The degree of squamous metaplasia (nucleo/cytoplasmic ratio) was evaluated following the scale of Nelson [39], and the density of goblet cells was calculated in relation to the area of the epithelial sheet and shown as cells/mm^2^.

The same battery of tests was repeated after 6 months.

#### 2.3.10. Statistical Analysis

The data were subjected to analysis using GraphPad Prism 8 for MacOS (GraphPad Software, San Diego, CA, USA). Descriptive statistics were applied to demographic variables and the presence of ocular conditions. The normality of the sample was assessed using the Shapiro–Wilk test. To compare quantitative variables before and after treatment, Student’s *t*-test for paired samples was employed when normality assumptions were satisfied. Alternatively, the Wilcoxon test was utilized when the data did not meet normality assumptions. For categorical variables, the Chi-square test was used. A significance level of *p* < 0.05 was assumed to determine statistical significance.

## 3. Results

A total of 23 eyes from 12 patients were analyzed, including 5 males and 7 females. The average age was 45.33 ± 3.62 years. No cases of intolerance or PRGF-related side effects were reported during its application.

### 3.1. General Condition at Baseline

At the beginning of the study, all patients presented some degree of AAK and ocular dryness. The next most frequent complication, as can be seen in Table 1, was the presence of glaucoma. Among the twelve patients, six exhibited asymmetries in the degree of AAK between their eyes. Five patients had the same degree of AAK in both eyes, while one patient was one-eyed, with a prosthetic eye in the other. Of the 12 patients, only 2 continued their previous AS treatment. Table 2 shows the presence of complications according to the degree of keratopathy associated with aniridia.

### 3.2. Clinical Test Results after Treatment

The results of the various analyzed variables are presented in Table 3.

#### 3.2.1. Symptomatology

The symptoms exhibited a statistically significant reduction during the follow-up visit, as indicated by both the OSDI and SANDE questionnaires, as depicted in Figure 1. One patient did not perceive any difference between the previously administered AS and PRGF. Another patient, who had not received an AS treatment before, stated that there was no improvement with the use of PRGF. The remaining patients reported symptom improvement following the application of PRGF. One of the patients stated that PRGF had alleviated her discomfort to such an extent that it had positively transformed her quality of life.

#### 3.2.2. BCVA, Ocular Redness, and Damage

The values of the BCVA increased but did not reach statistical significance. A statistically significant change was observed in ocular redness, which decreased during the follow-up visit. Damage to the ocular surface was also reduced—assessed by corneal staining (refer to Figure 2).

#### 3.2.3. Effects on Tear Film

A trend towards an increase in the FBUT was observed without a statistically significant difference, as was the case for the increase in the Schirmer test. As for the degree of meibomian gland loss, no variation was observed. See Figure 3.

#### 3.2.4. Impression Cytology

Impression cytology specimens from 7 out of 12 patients, including 11 eyes, were analyzed to evaluate the efficacy of the PRGF treatment. At the beginning of the study, before the application of the treatment, specimens from congenital aniridia patients showed a significant decrease in the goblet cell density compared with the healthy controls (Figure 4A,B). In addition, epithelial cells were found to be altered in terms of the relation nucleus to the cytoplasm, indicating epithelial metaplasia. In some cases, grade 3 squamous metaplasia was found (Figure 4B). Some specimens showed cells isolated in the membrane without intercellular cohesion (Figure 4B). Also, four patients showed stratified epithelial structures (Figure 4C).

At the end of the treatment with PRGF, three eyes corresponding to two patients continued showing altered epithelial cells, lacking cell-to-cell contact in the impression cytology specimens, like those shown in Figure 4B. In addition, the nucleocytoplasmic ratio was higher than that in normal specimens, indicating metaplasia. One additional patient also showed pathologically altered goblet cells (Figure 5C). The remaining four patients (seven eyes) showed an increased number of goblet cells at the end of the treatment (Figure 5B) compared with the specimens obtained from the starting day collection (Figure 5A). This increase in goblet cell density was found to be like that in healthy epithelia (534.70 ± 42.72 cells/mm^2^ in treated patients vs. 558.60 ± 29.79 cells/mm^2^ in healthy individuals). The increase in goblet cell density between the pre (52.36 ± 12.47 cells/mm^2^) and post-treatment (534.70 ± 42.72 cells/mm^2^) periods was statistically significant (*p* < 0.01). In addition, the nucleocytoplasmic ratio in these 11 specimens was close to normal (graded 0), showing a dense and continuous epithelial sheet (Figure 5B).

## 4. Discussion

This study investigated the effects of PRGF in patients with congenital aniridia and dry eye disease.

Currently, there are several biological eye drops, derived either from serum or plasma. These can be obtained from the patient’s own blood—known as autologous samples—such as AS drops, platelet-rich plasma (PRP), and PRGF, or allogeneic samples, such as serum derived from umbilical cord blood (UCBS) and peripheral blood serum obtained from a healthy donor. It has been demonstrated that drops made from blood derivatives offer an advantage over conventional therapy in the treatment of the ocular surface, not only as a substitute for tears but also as regenerative therapy, where different growth factors and the metabolite profile may play a determining role [40,41,42,43,44]. Thus, AS contains higher concentrations of lysozyme, vitamin A, TGF-β, and fibronectin than natural tears, maintaining a similar pH [45,46], and, in turn, it has been described that UCBS contains a higher concentration of metabolites and at least two times higher concentrations of EGF and TGF-A compared to AS, which is reported to provide a therapeutic advantage over AS [40,47]. However, UCBS—as an allogeneic sample—has the main disadvantage of potentially transmitting an infectious disease from the pregnant mother, so laboratory testing is required for its detection.

The main problem with blood-derived serums is the lack of international consensus on their preparation method. The different available formulations show inconsistency regarding the amount of blood extracted, centrifugation force, activator, and heating. In addition, there are inter-individual differences in platelet numbers as well as in the concentrations of different growth factors in PRP. However, it is known that with age, there are decreases in IGF and VEGF [48,49]. 

The use of PRGF as a treatment for various ocular disorders, including evaporative dry eye and epithelial disorders, among others, has been previously described [15,21,26,28,50,51,52,53,54,55]. In our work, we used PRGF Endoret^®^, which provides some significant advantages over other procedures, such as the use of 3.8% sodium citrate as an anticoagulant, which does not alter membrane receptors as EDTA would. Additionally, other preparations use acidic citrate, which is more acidic and may affect platelet aggregation. Platelet activation is achieved using calcium chloride, as it has shown a greater release of growth factors and a greater effect on the growth of corneal epithelial cells [44,56]. The centrifugation force is lower than that used to obtain AS or other PRPs [57], and its subsequent heating at 56° for one hour eliminates IgE and complement activity while preserving the concentration and activity of proteins, including growth factors (PDGF, TGF-β1, IGF, and VEGF) and fibronectin.

Focusing on aniridia, López-García et al. conducted a study involving AS, which was administered to 13 patients with AAK over a period of 2 months. The study concluded that the use of AS led to an improvement in symptoms for all participants. Corneal epithelialization, corneal squamous metaplasia, and tear stability significantly improved after treatment. However, only minimal improvements were observed in visual acuity, corneal vascularization, or subepithelial scarring [13].

As of the current date, an ongoing clinical trial (NCT05400590) is actively comparing the use of PRGF versus AS in patients with congenital aniridia. The trial’s communication date was in June 2022, with an estimated completion date of November 2025 [58].

In the present study, when analyzing the improvement of ocular discomfort symptoms after the use of PRGF, a statistically significant enhancement was observed for all of the analyzed questionnaires, including both the OSDI and SANDE in terms of frequency and severity. The initial average OSDI score was 30.54, which is categorized as a moderate level of impairment [59]. According to Miller K’s et al. work [60] for this level of impairment, a change of 4.5 to 7.3 was required to achieve a Minimal Clinically Important Difference (MCID). In this study, the patients reported an average improvement of 12.37, which can be interpreted as not only statistically significant but also clinically meaningful.

In relation to the frequency and severity of symptoms assessed using the SANDE questionnaire, a statistically significant improvement was also observed. Bahreini M et al. determined that a mean change of 16.55 mm in the questionnaire was necessary to perceive a change in pain severity [61]. Todd et al. determined that this change should be around 13 mm [62]. Our results showed a mean change of 30 mm in frequency and 28.75 mm in severity, which can be considered clinically significant.

Visual acuity in patients with congenital aniridia is typically low, often around 0.19 on the decimal scale [6]. The average visual acuity of our sample aligned with this value.

Based on the obtained data, there was minimal variation in the BCVA following the PRGF treatment, and no statistically significant differences were observed at the follow-up visit. Drawing a direct analogy with patients solely diagnosed with dry eye disease and treated with PRGF is not straightforward in the case of congenital aniridia. There are numerous factors that can affect BCVA in patients with congenital aniridia. These factors may include foveal or optic nerve hypoplasia, nystagmus, photophobia, cataracts, or aniridic keratopathy [9].

Investigations into the tear fluid of patients with aniridia have shown significantly elevated levels of various proinflammatory cytokines, such as IL-1, IL-6, IL-17, and IL-23, among others, like what occurs in DED [63]. In line with this, our team recently published a study demonstrating a reduction in ocular surface inflammation, assessed by ocular redness, in patients diagnosed with DED and treated with PRGF [25]. Now, in this study, a statistically significant decrease in ocular redness was again observed after PRGF treatment, indicating a reduction in ocular inflammation. This is noteworthy, as ocular redness is considered a primary clinical sign of ocular inflammation [32,33,64]. The observed reduction in ocular redness in PRGF-treated patients could be attributed to the anti-inflammatory properties of TGF-β present in PRGF [16,65].

Following PRGF application, a statistically significant improvement in ocular surface damage, assessed through corneal fluorescein staining, was also observed. This reduction in ocular surface damage was further supported by impression cytology, where four out of seven patients studied experienced a significant increase in the number of goblet cells. In addition, the morphology of the epithelial sheet was restored, and the presence of stratified structures and mucous accumulations was low, suggesting a recovery of the conjunctiva surface integrity and function. These features have been related to an improvement in dry eye disease symptoms [66].

The tear film exhibited a tendency towards stability, as indicated by the increased mean values of FBUT, although no statistical significance was obtained. Nevertheless, the final values remained well below the 10 s threshold, which is considered normal [67]. The results of the Schirmer test demonstrated a statistically significant increase.

No changes were observed in terms of the degree of Meibomian gland loss after the PRGF treatment during the analyzed period. However, it is worth noting that the values obtained from the initial visit already indicated a moderate level of gland loss, despite the relatively young ages of the studied patients. These findings are in line with previous research that demonstrated an increased degree of Meibomian gland loss, heightened Meibomian gland dysfunction (MGD), and decreased expressibility in patients with congenital aniridia compared to a control group, suggesting that MGD is involved in the pathogenesis of DED in aniridia [2].

This study has several limitations, including a small sample size, although it is important to consider that aniridia is classified as a rare disease with a low incidence. Additionally, a similar number of eyes were studied compared to other similar published works [13]. Another limitation is that there was no subdivision according to the type of DED—evaporative or secretion deficit—or according to the severity of symptoms [68,69]. The loss of corneal sensitivity and the role of corneal innervation described by other authors [2] were not analyzed either due to the difficulty encountered due to the presence of nystagmus.

Measuring different variables in patients with congenital aniridia is often quite complex. For instance, assessing the FBUT requires intense illumination, which can cause significant discomfort for these patients who lack an iris and often experience photophobia. Holding the eyelids for proper evaluation becomes necessary, and reflex tearing due to photophobia-induced glare is frequent. This situation might lead us to consider discarding the FBUT and using the NIBUT (non-invasive break-up time) instead. However, we encounter another challenge because a significant number of patients with congenital aniridia also have nystagmus and significant corneal deformation due to aniridic keratopathy, making an accurate computerized analysis difficult. Nystagmus also complicates the sampling process during impression cytology. Therefore, in our study, we decided to collect only conjunctival samples and avoid approaching the corneal area to prevent corneal damage.

As previously mentioned, symptoms of ocular dryness are prevalent in patients with congenital aniridia. When using tools designed to assess DED, they may not perfectly align with the profiles of these patients. For example, some studies in the literature describe using the OSDI questionnaire but excluding questions related to visual acuity [70]. In our study, we decided to include those questions because while the BCVA of some participants was too low for activities like driving or computer work, others still maintained a VA of 0.5, which is sufficient for reading or watching television.

It is also important to consider that glaucoma is frequently present in congenital aniridia—in our study, nearly half of the patients had glaucoma. The use of topical glaucoma medications is common and known to cause redness of the ocular surface and eyelids as well as irritation [71,72,73]. In our study, we decided not to intervene in this regard and analyzed each patient without modifying their prescribed intra ocular pressure (IOP) control treatment, as they would continue to maintain the same treatment regimen. The IOP values remained unchanged from those obtained at baseline, so it does not appear that PRGF treatment had any influence on IOP.

## 5. Conclusions

This study provides an analysis of different variables when using PRGF in patients with congenital aniridia and symptoms of ocular dryness. It can be concluded that the use of PRGF resulted in a significant improvement in symptoms, ocular inflammation, and ocular damage. No adverse effects were observed during the use of PRGF.

## Figures and Tables

**Figure 1 diseases-12-00076-f001:**
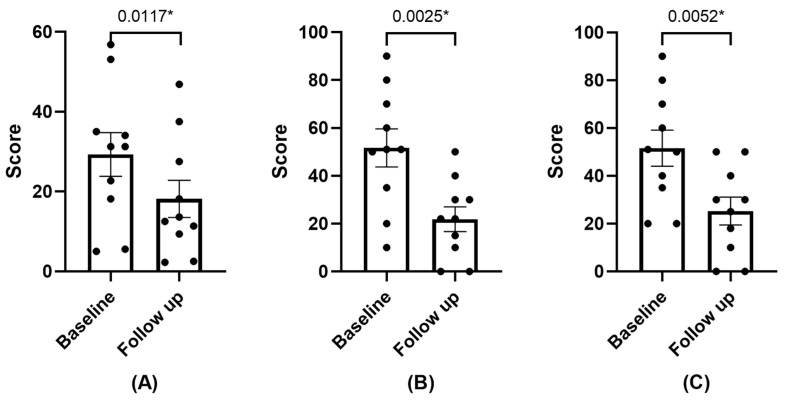
Mean and SEM of OSDI (**A**), SANDE Frequency (**B**), and SANDE Severity (**C**) questionaries at baseline and follow-up visit. * Statistically significant difference (*p* < 0.05).

**Figure 2 diseases-12-00076-f002:**
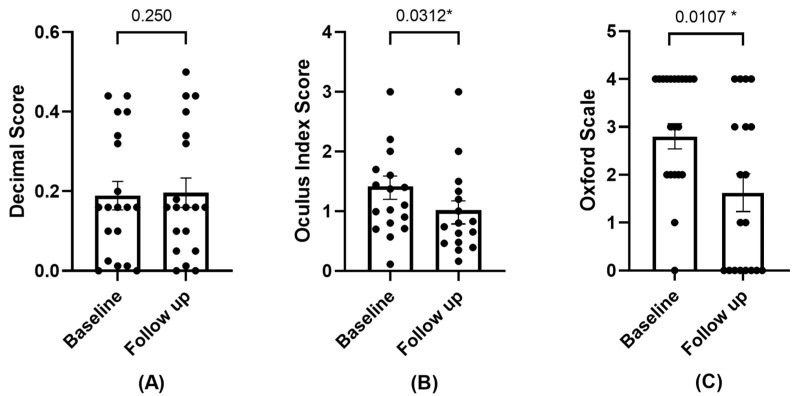
Mean and SEM of BCVA (**A**), ocular redness (**B**), and corneal staining (**C**) at baseline and follow-up visit. * Statistically significant difference (*p* < 0.05).

**Figure 3 diseases-12-00076-f003:**
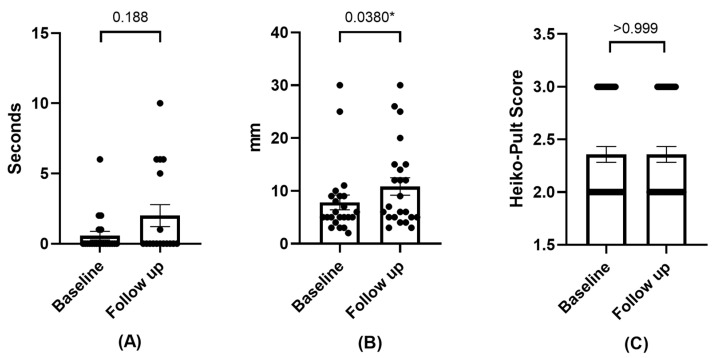
Mean and SEM of FBUT (**A**), Schirmer test (**B**), and meibography (**C**) at baseline and follow-up visit. * Statistically significant difference (*p* < 0.05).

**Figure 4 diseases-12-00076-f004:**
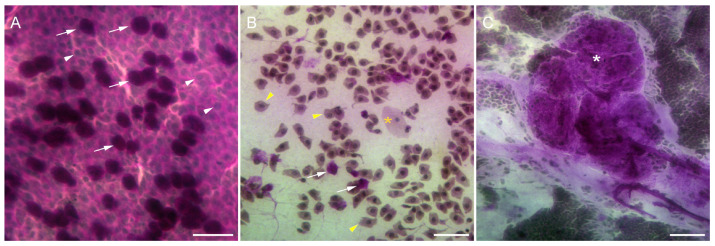
Impression cytology samples from (**A**) a healthy volunteer, showing a high number of goblet cells (arrows) and a large sheet of epithelial cells closely attached with a normal nucleocytoplasmic ratio (arrowheads); (**B**) a patient suffering from aniridia showing severe epithelial metaplasia (yellow arrowheads) and a very low number of goblet cells (arrows). The yellow asterisk marks an extra large metaplasic epithelial cell. Image (**C**) shows a detail of a stratified epithelial structure (asterisk) obtained from the conjunctiva of a patient with aniridia. (Scale bars: 50 µm in (**A**,**B**) and 100 µm in (**C**).)

**Figure 5 diseases-12-00076-f005:**
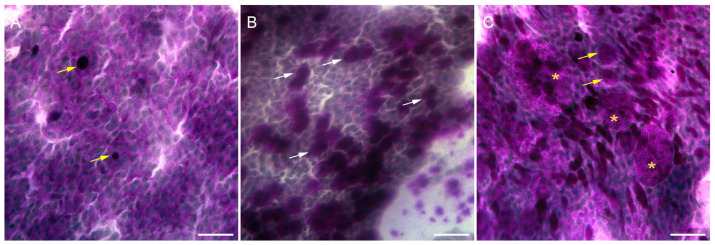
Impression cytology samples from (**A**) an untreated patient showing atrophic and scarce goblet cells (yellow arrows); (**B**) a patient after 6 months of treatment with PRGF showing a recovered number of goblet cells (arrows); and (**C**) a detail of the sample from one patient experiencing alterations in the cellular morphology of goblet cells (yellow arrows) and showing giant cells, possibly coincident with inflammatory mast cells (yellow asterisks). (Scale bars: 50 µm.)

**Table 1 diseases-12-00076-t001:** The presence of complications at the baseline visit, reported in the number of eyes affected, and prior to treatment using an autologous serum.

AAK	Cataract	Glaucoma	Nystagmus	Photophobia	Transplant	Dryness	AS
23 (100%)	18 (78.3%)	10 (43.5%)	13 (56.5%)	11 (47.8%)	4 (17.4%)	23 (100%)	9 (39.1%)

Aniridia-associated keratopathy (AAK), autologous serum (AS).

**Table 2 diseases-12-00076-t002:** The presence of complications at the baseline based on the degree of aniridia-associated keratopathy.

AAK	*N* (RE:LE)	Cataract	Glaucoma	Nystagmus	Photophobia	Transplant	Dryness	AS
1	4:3	6 (85.7%)	5 (71.4%)	5 (71.4%)	2 (28.6%)	0 (0%)	7 (100%)	2 (28.6%)
2	1:4	5 (100%)	5 (100%)	3 (60%)	2 (40%)	0 (0%)	5 (100%)	2 (40%)
3	5:0	4 (80%)	3 (60%)	1 (20%)	5 (100%)	1 (20%)	5 (100%)	2 (40%)
4	2:4	3 (50%)	2 (33.3%)	4 (66.7%)	2 (33.33%)	3 (50%)	6 (100%)	3 (50%)

Aniridia-associated keratopathy (AAK), right eye (RE), left eye (LE), autologous serum (AS).

**Table 3 diseases-12-00076-t003:** Overall results expressed as mean ± standard error of mean.

	Baseline	Follow-Up	*p*-Value
OSDI (score)	30.54 ± 5.98	18.17 ± 5.22	0.01 *
SANDE Frequency (score)	51.88 ± 10.09	21.88 ± 18.50	0.003 *
SANDE Severity (score)	51.88 ± 9.54	23.13 ± 6.47	0.005 *
BCVA (decimal)	0.19 ± 0.04	0.20 ± 0.04	0.25
Ocular Redness Index	1.47 ± 0.29	0.93 ± 0.31	0.03 *
Corneal Fluo Staining (Oxford)	2.82 ± 0.30	1.82 ± 0.40	0.01 *
FBUT (s)	0.73 ± 0.42	2.27 ± 0.86	0.19
Schirmer (mm)	7.83 ± 1.40	10.83 ± 1.65	0.04 *
Superior Meibography (score)	2.37 ± 0.11	2.42 ± 0.12	>0.99
Inferior Meibography (score)	2.40 ± 0.11	2.35 ± 0.11	>0.99

Ocular Surface Disease Index (OSDI), Symptoms Analysis in Dry Eye (SANDE), best corrected visual acuity (BCVA), fluorescein break-up time (FBUT). * Statistically significant difference (*p* < 0.05).

## Data Availability

The data used to support this study’s findings are available by contacting the corresponding author upon reasonable request.

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
