# Peer review of "Efficacy and Safety of Plasma Rich in Growth Factor in Patients with Congenital Aniridia and Dry Eye Disease"

_diseases, 2024, doi:10.3390/diseases12040076_

Round 1

Reviewer 1 Report

Comments and Suggestions for Authors

11)    Results section:

Please modify all the graphs with bars and dots, with dots representing values for each single patient

Please modify figure 4 and 5 adding white arrows indicating what described in the figure caption.

22) Discussion section:

Please add to the discussion the following arguments and citations:

-Discuss the fact that also metabolites can have a role in improving tear films

Please refer to

Quartieri E, Marraccini C, Merolle L, Pulcini S, Buzzi M, Guardi M, Schiroli D, Baricchi R, Pertinhez TA. Metabolomics comparison of cord and peripheral blood-derived serum eye drops for the treatment of dry eye disease. Transfus Apher Sci. 2021 Aug;60(4):103155. doi: 10.1016/j.transci.2021.103155. Epub 2021 May 8. PMID: 33975808.

-Discuss briefly which are the technical differences with other similar preparations

-Discuss which are the limitations of the study relatively to the clinical tests used herein

Please refer to

 Wolffsohn JS, Travé Huarte S, Jones L, Craig JP, Wang MTM; TFOS ambassadors. Clinical practice patterns in the management of dry eye disease: A TFOS international survey. Ocul Surf. 2021 Jul;21:78-86. doi: 10.1016/j.jtos.2021.04.011

-Discuss the inter-donors variability in the compositions of GFs (as it was previously published) and discuss how it is necessary to standardize the production of PRP and of PRGF as well.

Reviewer 2 Report

Comments and Suggestions for Authors

I appreciate the opportunity to review this interesting report on safety and efficacy of PRGF in patients with eye diseases. However, the findings by the authors are poorly represented here in the manuscript and lots of improvement and corrections needed in the manuscript before its publication.

Introduction needs to be rewritten. Some places authors used Isolated aniridia, Hereditary aniridia. How they differ from each other.

Line 37-38: Its prevalence is 1-9/100,000, and its annual incidence is estimated to be 1/64,000-1/96,000. If the author can mention percentage or in words like one in million etc.

Materials and Methods section: This section is jumbled and not organized well. No number assigned to subheadings for various methods applied.

Results section: The results are not represented in order wise and no separate section given within the results. They are all continuous.  No subheadings mentioned.

The figure should be with legends. They are separate. Also, the legends are not depicting properly the results figure. The graph in Fig 2, the y-axis reveals incomplete information. In Fig 3A, y-axis no labeling. In Fig 4, there can be some arrows drawn within that can represent the figure legends. Same goes for Fig 5.

Discussion: There are too many separate paragraphs. If they can be merged into two or three paragraphs.

Comments on the Quality of English Language

The minor English language editing needed and proofreading of manuscript is required. 

Reviewer 3 Report

Comments and Suggestions for Authors

The paper „Efficacy and Safety of Plasma Rich in Growth Factor in Patients with Congenital Aniridia and Dry Eye Symptoms“ is appropriate for the Journal Diseases and within the scope.

Some moderate issues should be addressed before publication such as:

1. In the abstract the first sentence starts with lower case instead of Upper Case.

2. Please add after line 105-106 what were non-inclusion criteria. Moreover it should be stated that the patients could withdraw their consent at any given point and that their further treatment would not be affected if this is the case.

3. Line 266, the title Figure 1 should be below the figure itself and to be below the figure with the capitation: Figure 1. Mean and SEM of OSDI (A), SANDE Frequency (B) and SANDE Severity (C) questionnaires. *Statistically significant difference (p<0.05).

Line 275, the same comment as Figure 1.

Line 282, the same comment as Figure 1.

Line 300, the same comment as Figure 1.

4. In Conclusion please add that PRGF did not affect the effectiveness of the medication applied for their prescribed ocular pressure control treatment as that seems to be the point of lines 438-439 or even other medication applied for eye complications.

5. In addition to that since in line 245 is given in the Table that there were other complications such as cataract, glaucoma, nystagmus, photophobia, transplant, dryness etc. please provide data whether patients were treated with some medications in order to point the lack of interactions of PRGF as stated in lines 438-49.

Round 2

Reviewer 2 Report

Comments and Suggestions for Authors

I think that the authors have adequately addressed the comments made by me in the revised version of the manuscript. Though some minor corrections needed in the revised version before acceptance. 

Line 44: Check -so, no-isolated. I didn't get this. 

Figure 2B: Check "y" font size.

Check the tables 1, 2 and 3. Not aligned well. Also, if each table have an outside border or are in boxes. 
